# Dairy Calf Welfare and Factors Associated with Diarrhea and Respiratory Disease among Chilean Dairy Farms

**DOI:** 10.3390/ani10071115

**Published:** 2020-06-29

**Authors:** Javiera Calderón-Amor, Carmen Gallo

**Affiliations:** 1Escuela de Graduados, Facultad de Ciencias Veterinarias, Universidad Austral de Chile, Valdivia 5090000, Chile; 2Instituto de Ciencia Animal, Facultad de Ciencias Veterinarias, OIE Collaborating Centre for Animal Welfare and Livestock Production Systems, Universidad Austral de Chile, Valdivia 5090000, Chile; cgallo@uach.cl

**Keywords:** dairy calf, animal welfare, health, diarrhea, respiratory disease

## Abstract

**Simple Summary:**

Since 2013, a Chilean law regulates the welfare of farm animals. Despite advances in scientific knowledge and legislation, many farms use management practices that can be detrimental to animals. The objectives of this study were to describe common management practices that may affect the welfare status of unweaned dairy calves and identify factors associated with diarrhea and respiratory disease. We visited 29 dairy farms and collected information on management practices, environment, and animal health. Management practices identified as risk factors for poor calf welfare were: reliance on the mother to provide colostrum, use of restrictive milk feeding (<4 L/day), disbudding performed with no pain control, and lack of appropriate euthanasia protocols. Factors associated with diarrhea were: frequency of bed cleaning, calf cleanliness score, type of milk, and herd size. Factors associated with respiratory disease were: calf cleanliness score, pen space allowance, and colostrum quality evaluation. Showing critical points that impact the health and welfare of dairy calves facilitates the implementation of improvement strategies.

**Abstract:**

This study aimed to describe management practices that may compromise the welfare of unweaned dairy calves on 29 dairy farms in Chile, and identify factors associated with diarrhea and respiratory disease (n = 700 calves). Evaluations were divided into protocol-, facility-, and animal-based measurements. Calf diarrhea and respiratory disease data were analyzed using logistic regression models. Management practices identified as risk factors for poor calf welfare were: relying on the mother to provide colostrum (48.0% of the farms); using restrictive milk feeding (65.5%), and unpasteurized waste milk (51.7%); giving water after 30 days of age (17.2%); disbudding without analgesia (89.6%) or anesthesia (79.3%); lacking euthanasia protocols (61.5%). Factors significantly (*p* < 0.05) associated with increased odds of diarrhea were: cleaning the calves’ bed once a week and 2–3 times a week compared with every day, using milk replacer and untreated waste milk compared with treated waste milk (pasteurized or acidified), animals scored dirty in the calf cleanliness score compared with clean animals, and greater herd size. Factors significantly associated with increased odds of respiratory disease were: less pen space allowance (<1.8 m^2^), farms that did not check colostrum quality, and animals that scored dirty and moderately dirty compared with clean calves. These results suggest the need to improve specific management practices associated with reduced welfare and health in dairy calves in Chile.

## 1. Introduction

The concept of animal welfare indicates the state of an individual and the way it copes with its environment [1]; it encompasses the physical welfare and the mental state of a sentient animal [2]. The rearing period of dairy calves is critical for animal welfare reasons [3], and the future productive value of the animals [4]. Unlike other productive systems in dairy production, the newborn is separated from the mother at birth, or close to it, and then, raised artificially. These animals often face challenges such as isolation, milk restriction, and weaning at an early age [3,5,6].

According to the European Welfare Quality^®^ protocol [7], four general criteria must be evaluated on-farm to define the level of welfare in dairy calves: good nutrition, good housing, appropriate behavior, and good health. Good nutrition refers to an adequate diet for maintenance and growth; a good practice from behavior and productive point of view is to provide higher amounts of milk or milk ad libitum [8,9]. Good housing relates to an appropriate environment for animals, considering thermal, physical, psychological, and behavioral comfort [10]. Appropriate behavior refers to the expression of positive and social behavior; housing calves in group pens is associated with higher solid feed consumption, weight gain, and more play behavior [6,11,12]. Good health refers to the absence of illness and pain produced by routine management procedures; the two most important health problems that affect dairy calves’ welfare are diarrhea and respiratory disease. Windeyer et al. [13] described an incidence of 23% for diarrhea and 22% for respiratory disease among Canadian dairy farms. Both are multifactorial diseases and are linked to different management practices. Medrano-Galarza et al. [14] reported that adding more frequently fresh bedding was associated with a lower prevalence of diarrhea, and feeding whole milk instead of milk replacer was associated with a lower prevalence of respiratory disease. 

Few data are available regarding the rearing conditions of dairy calves from the animal welfare point of view in Chile. Therefore, this study’s objectives were to describe the most common management practices discussing the critical points associated with reduced welfare status in unweaned dairy calves, and investigate the associations of management and environmental factors with diarrhea and respiratory disease. 

## 2. Materials and Methods 

The Animal Care Committee and Bioethics Committee in Humans from Universidad Austral de Chile approved the experimental protocol under permission No. 359 and Ord. 287, respectively.

We used a convenience (non-random) sample of 31 dairy farms distributed throughout Los Ríos region, Chile, during September 2017 and July 2018. This region has the second-largest number of dairy cattle of the country (27.7% of the national population), with approximately 736 small- and large-scale dairy farms [15,16], which are fundamentally pasture-based systems [17]. The number of farms included was determined predominantly by time and budget constraints associated with data collection and was the maximum number that could be recruited and assessed within the study period. Two selection criteria were established: use of artificial rearing system of calves-separation from the dam before seven days postpartum- and the use of group pens at some point during the preweaning period. We excluded two farms from the data due to the lack of accuracy in the responses and missing information (total farms = 29). The measurements were classified as protocol-, facility-, and animal-based. 

### 2.1. Protocol-Based Measurements 

We developed a survey based on Vasseur et al. [3] (Appendix A. Calf management survey applied on 29 dairy farms using group pens in Los Ríos region, Chile) using closed and open-ended questions. The survey was collected through a face-to-face interview with the calf manager. It was divided into six sections: general, calving management, colostrum management, calf feeding, weaning, and painful procedures.

### 2.2. Facility-Based Measurements 

We recorded the following housing conditions from 79 pens: time in individual hutches, type of bedding material, straw depth, space allowance, and bed cleaning frequency. Time in individual hutches was asked directly to the calf manager. Type of bedding material was measured by visual inspection. Straw depth was assessed once using a measuring tape in the middle of the resting area. Space allowance was evaluated using a measuring tape to register the length and width of the pen; two farms could not be assessed because calves were kept on pasture in large paddocks. Bed cleaning frequency was asked directly to the calf manager in the form of “How often do you remove wet, dirty bedding and replace it with clean bedding?”

### 2.3. Animal-Based Measurements 

The measurements were carried out only once per calf by the same researcher with veterinary training (female, 1.58 m high, blue coverall). Only unweaned calves in group pens were considered for evaluation. The number of evaluated animals per farm depended on the total number of calves present at the time of the visit: if there were fewer than 30 calves, all of them were evaluated; if there were 30 or more calves, pens were chosen on a pseudo-random basis until approximately 30 measurements were reached (total = 700 calves). Additionally, the sex and date of birth of each calf were collected.

The health assessment included a visual evaluation inside the pens. The health chart included: body condition score, calf cleanliness score, fecal score, eye score, cough, nasal discharge, skin lesions, lameness, and other health events (not considered in the chart). All evaluations were based on the European Welfare Quality^®^ protocol [7] and the protocol of the University of Wisconsin-Madison [18]. Table 1 summarizes the health criteria and scoring system. A calf was diagnosed with diarrhea when it had a fecal score of 2 or 3. Fourteen animals could not be evaluated for diarrhea (total = 686 calves). A calf was diagnosed with respiratory disease when it presented cough score 2 or 3, and nasal discharge (score 1, 2, or 3) or ocular discharge (score 1, 2, or 3).

### 2.4. Editing of Data and Statistical Analysis

The classification of the variables was based on the distribution of the responses, common management practices, scientific knowledge, and quartiles.

The presence of a maternity pen, supervision during calving, colostrum quality evaluation, passive transfer evaluation, use of analgesia during disbudding, use of anesthesia during disbudding, and euthanasia protocol were coded as binary (yes/no). Age of the calves (days), time that mother and calf spent together (hours), number of calves in the calf barn, age of access to water, starter, and hay (days), time in the individual hutches (days), and bed material depth (cm) were used as continuous variables. For the variable season of birth, dates from 23 September to 21 December were defined as spring, from 21 December to 20 March as summer, from 20 March to 21 June as autumn, and from 21 June to 23 September as winter. Sex was classified as female and male. Breed was coded as Holstein purebred, dairy and beef crossbreeds, Holstein and Jersey crossbreed. Colostrum source was categorized as manual (use of bottle or tube feeding) and mother (colostrum received directly from the mother). Milk type was grouped as milk replacer, treated waste milk (pasteurized or acidified waste milk), and untreated waste milk. The liters of milk variable was coded as ≤4 L/day and >4 L/day. Milk feeding frequency was classified as 2 times/day and 4 times/day. Bed cleaning frequency was classified as every day, 2–3 times/week, and 1 time/week. Space allowance was first calculated per pen, and further classified as ≤1.8 m^2^/calf and >1.8 m^2^/calf according to the legal requirement in Europe [19]; the two farms that kept the calves on pasture were classified as >1.8 m^2^/calf. Type of bedding material was classified as straw, pasture, or without bedding material. Body condition was classified as normal condition, lower condition, and severe lower condition. Calf cleanliness score was classified as clean, moderately dirty, and dirty. All health conditions were coded as binary (healthy/sick), according to the criteria described above (Table 1). 

The data were stored in an Excel spreadsheet (Microsoft Office Excel^®^ 2016) and for further statistical analysis, were exported to SAS (version 9.4, SAS Institute, Cary, NC, USA). A statistical significance level of *p* < 0.05 was established. The results regarding protocol-, facility, and animal-based measurements were first summarized through descriptive statistics (mean, standard deviation, percentages, minimum and maximum values). 

Within-herd prevalence of diarrhea and respiratory disease were calculated using SUMMARY procedure in SAS. Respiratory disease and diarrhea data were analyzed using a multilevel logistic regression model for each health condition (PROC GLIMMIX in SAS). The predictor variables were selected from the protocol-, facility-, and animal-based measurements. First, univariate associations between the dependent variable and the potential predictors were examined one at a time. Pen nested within the farm was included as a residual-side random component. Continuous variables were categorized based on quartiles when the linearity assumption was not fulfilled; the variable N° of calves in the calf barn was the only one categorized into quartiles. Potential explanatory variables were assessed for collinearity (PROC CORR in SAS); if the correlation was >0.6, the variable with the highest *p*-value in the univariate analysis was not included in further analysis.

Second, variables with *p*-values ≤ 0.2 in the univariate analysis were selected for the multivariable analysis. The models were built by the manual backward selection procedure, controlling pen nested within the farm as a random effect, using 10 adaptive Gauss–Hermite quadrature points. The process continued until all variables were significant. Potential confounders were kept in the model by considering changes in coefficients greater than 30%. Biologically relevant interactions were tested; no significant interactions were found (*p*-value > 0.05). Outliers were examined graphically by plotting residuals. In both models, the removal of extreme values did not affect the results; therefore, we decided to keep the outliers in the final models. Odds ratios were calculated from the final parameter estimates. The models were assessed for goodness-of-fit using the receiver operating characteristic (ROC) curve.

## 3. Results

Table 2 summarizes the results of, protocol-, facility-, and animal-based measurements.

### 3.1. Protocol-Based Measurements

#### 3.1.1. General

The average number of lactating cows was 444 ± 259 (min = 48; max = 1200). The mean number of calves (unweaned and weaned) present in the calf barn at the time of the visit was 165 ± 136 (min = 20; max = 585). Thirty-eight percent of the farms had less than 71 calves in the calf barn; 20.7% had between > 71 to ≤ 129; 24.1% had > 129 to ≤ 215; the rest had more than 215 calves. The distribution of breeds was Holstein purebred (41.4%); dairy and beef crossbreeds (27.6%); Holstein and Jersey crossbreed (31.0%).

#### 3.1.2. Calving Management

Thirty-four percent of the farms did not have a maternity pen for calving, and 3.6% of them did not have farm staff for calving supervision. The average time that mother and calf spent together was 23.1 ± 31.9 hours (min = 0; max = 120).

#### 3.1.3. Colostrum Management

Forty-eight percent of the farms kept the mother with the calf for more than 24 hours, and thus, in all these farms, colostrum consumption was not controlled. The rest of the farms used bottles (44.8%) or tube feeding (6.9%) to ensure the consumption of colostrum. Evaluation of colostrum quality (use of colostrometer or refractometer) and passive transfer was not performed in 68.9% and 72.4% of the farms, respectively.

#### 3.1.4. Calf Feeding

Untreated waste milk was used in 51.7% of the farms, 27.6% used milk replacer, and 20.7% treated waste milk. Most farms (65.5%) used a restrictive feeding system (4 liters of milk per day) delivered twice a day (96.6%). The average age of access to water was 10.3 ± 15.5 days (min = 1, max = 90); 41.4% of the farms gave water from birth, and 17.2% provided water 30 days after birth. The average age of access to concentrate was 6.3 ± 8.3 days (min = 1; max = 30); 48.3% of the farms gave concentrate from birth, and two farms provided concentrate 30 days after birth. The average age of access to hay was 14.9 ± 16.9 days (min = 1; max = 90); 20.7% of the farms gave hay from birth, and 24.1% provided hay 30 days after birth.

#### 3.1.5. Weaning

Most farms used age as a criterion for weaning, with a mean age of 79.8 ± 18.0 days (min = 45, max = 120); the rest of the farms used the weight of the animal, with an average of 79.4 ± 14.8 kg (min = 60, max = 100).

#### 3.1.6. Painful Procedures

All farms performed disbudding with an average age of 44.9 ± 28.3 days (min = 15, max = 120), and 31.0% of the farms performed the procedure after two months of age. Most of the farms did not use analgesia (89.7%) or anesthesia (79.3%) during disbudding. Regarding euthanasia, 61.5% of the farms did not perform it; of those farms, 11.5% bled the calves without previous stunning, and 50.0% let the animals die in the pens (unassisted death).

### 3.2. Facility-Based Measurements

After mother–calf separation, 31.0% of the surveyed farms had a mixed system, in which calves were initially kept for a time in individual hutches. The average time the calves remained in individual hutches was 6.4 ± 4.7 days (min = 2; max = 20); on two farms, calves spent more than two weeks housed individually.

Most farms used straw as bedding material (86.2%), with an average of 7.9 ± 4.4 (min = 0, max = 20) cm of depth in the resting area; in 27.8% of the pens, the straw depth was less than 5 cm. In 6.9% of the farms, calves were kept on pasture, and 6.9% did not use bedding material, keeping the animals in slotted wooden floor systems during the entire preweaning period.

The average space allowance in the pens was 2.5 ± 0.9 m^2^/calf (min = 1.3; max = 6.5). In 29.1% of the evaluated pens, calves were kept with less space than recommended (1.8 m^2^/calf), and on eight farms, more than 50% of the calves were maintained in less space than 1.8 m^2^/calf.

Most of the calf managers answered that they cleaned the calves’ bed every day (44.4%), 40.7% two to three times a week, and 14.8% once a week.

### 3.3. Animal-Based Measurements

The mean number of recruited calves per farm was 24 ± 7 (total calves = 700). An average of 3 ± 1 pens per farm was evaluated, and the mean number of calves per pen was 13 ± 8. The average calf age was 45 ± 20 days, and most were females (63.3%).

According to the body condition score, 85.1% of the evaluated animals were classified as normal, 12.7% as lower body condition, and 2.2% as severe lower body condition. On most farms (n = 26), at least one calf was scored as lower or severe lower body condition, and on one farm, more than 50% of the animals had lower or severe lower body condition. According to the calf cleanliness score, 71.3% of the calves were classified as clean, 25.4% as moderately dirty, and 3.3% as dirty. On most farms (n = 22), at least one calf was scored as dirty or moderately dirty; and on six farms, more than 50% of the evaluated calves were scored as dirty or moderately dirty. 

Table 3 shows calf and within-herd prevalence of respiratory disease, diarrhea, and other health problems. As a summary of health status, all farms presented at least one animal with a health problem, and a high percentage of animals (37.2%) were sick (with at least one health problem). On seven farms, more than 50% of the evaluated calves were sick. We found a prevalence of 15.5% and 13.7% of respiratory disease and diarrhea, respectively. On most farms, at least one calf was diagnosed with respiratory disease (n = 25 farms) or diarrhea (n = 22). Fourteen calves presented both health conditions. 

#### 3.3.1. Diarrhea Model

Table 4 shows the final model with factors associated with diarrhea. A lower number of calves in the calf barn was associated with decreased odds of having diarrhea. Cleaning the calves’ bed once a week or two to three times a week compared with cleaning every day, calves scored moderately dirty in the calf cleanliness score compared with clean score calves, and the use of milk replacer or waste milk compared with treated milk had greater odds of diarrhea.

#### 3.3.2. Respiratory Disease Model

Table 5 shows the final model with factors associated with respiratory disease. Farms that did not check the quality of colostrum, calves that scored moderately dirty and dirty in the calf cleanliness score compared with clean score, and pens with lower space allowance (≤1.8 m^2^/calf) were associated with increased odds of respiratory disease. The frequency of bed cleaning was not related to respiratory disease, although this variable was kept in the model to control for confounding.

## 4. Discussion

Our results indicate that several management practices affect the welfare of calves in Chile; likewise, we found protocol-, facility-, and animal-based factors significantly associated with diarrhea and respiratory disease.

### 4.1. Protocol-Based Measurements

#### 4.1.1. General

The total number of calves in the calf barn was associated with diarrhea. Herds with ≤ 71 calves [OR= 0.85; *p*_wald_ <0.05; CI_95_ = 0.02, 0.33] and herds with >129 to ≤ 215 calves [OR= 0.74; *p*_wald_ < 0.05; CI_95_ = 0.02, 0.43] (compared with > 215 calves) were associated with decreased odds of diarrhea. Herd size has been described in other studies as a risk factor for diarrhea [20] and mortality in calves [21,22]. Klein-Jöbstl et al. [20] explained that the association between diarrhea and greater herd size could be due to the insufficient caretakers on large-scale farms. Another explanation could be that larger herds may enhance the transmission of pathogens among animals, especially because our variable considered both weaned and unweaned animals within the calf barn.

#### 4.1.2. Calving Management

The current study found that 34.0% of the farms did not have a maternity pen for calving; therefore, cows gave birth in the same paddocks where they were kept during the dry period. Vasseur et al. [3] reported that 51.3% of surveyed farms in Canada did not use a maternity pen, even if they had one. It is fundamental to provide cows with a particular infrastructure for calving; this helps to minimize stress and ensures comfort and hygiene, which is associated with a reduction in problems during calving and perinatal mortality [23]. It also facilitates the supervision of the farmers [3].

A high proportion of the farms allowed mother–calf contact for more than 24 h. Vasseur et al. [3] and USDA [24] found that only 7.8% and 7.3% of the surveyed farms kept the cow and the calf together, respectively. On the one hand, early separation is recommended since it decreases the risk of calves’ exposure to environmental pathogens and facilitates the consumption of colostrum [3]. Besides, separation after the mother–calf bond development brings negative consequences for the calves in terms of behavior and emotional state [25]. On the other hand, keeping the mother with the offspring has behavioral and productive advantages in calves, such as reducing cross-sucking and increasing weight gain [26]. 

#### 4.1.3. Colostrum Management

Almost half of the surveyed farms did not control colostrum consumption. This is a higher percentage compared with studies in Canada (15.6%) [3] and Brazil (40%) [27]. Relying on the mother, the consumption of colostrum could lead to failure in passive transfer. According to Franklin et al. [28], calves that ingested colostrum from their mothers had fewer serum proteins than calves that were manually fed colostrum. Feeding colostrum with a nipple bottle or esophageal tube seems to have similar passive transfer efficiency [29]. The esophageal tube is associated with higher production rates and improves glucose metabolism [29,30]. Still, regurgitation and inhalation are possible consequences in force-fed calves [31].

Although most farms did not measure colostrum quality, it was surprising that 31.04% did measure it. This result is similar to Pempek et al. [32], who also reported a low percentage of farms (either conventional or organic) controlling colostrum quality, and to Vasseur et al. [3], who reported none. We found that farms that did not evaluate colostrum quality were associated with increased odds of respiratory disease [OR = 2.28; *p*_wald_ < 0.05; CI_95_ = 1.01, 5.13]. Shivley et al. [33] found that poor colostrum quality (IgG ≤ 50 g/L) was a risk factor for passive transfer failure. Failure of passive transfer immunity (measured by serum total protein or γ-globulin concentration) has been recognized as a risk of calves’ respiratory disease [13,34]. Donovan et al. [35] reported that the risk of pneumonia increased at lower serum total protein. Hence, farms that perform colostrum quality management would be more successful in passive transfer.

Most farms in our study did not evaluate passive transfer of immunity as routine management. Vasseur et al. [3] reported that no surveyed farms checked passive transfer. This management helps to identify animals with low serum immunoglobulin levels and susceptible to disease [36]. Failure of passive transfer is associated with negative outcomes such as poor growth performance [37] and higher morbidity and mortality [35].

#### 4.1.4. Calf Feeding

A large number of farms fed their calves with untreated waste milk and milk replacer. The use of untreated waste milk reported in this study was higher than in the USA (31%) [24] and Brazil (35%) [27]. We found that the type of milk was associated with the presence of diarrhea: farms that fed calves with untreated waste milk [OR = 31.02; *p*_wald_ < 0.05; CI_95_ = 5.65, 170.21] or milk replacer [OR = 8.56; *p*_wald_ < 0.05; CI_95_ = 1.18, 61.94] had higher odds of having calves with diarrhea (compared with treated waste milk). Waste milk increases the transmission of infectious pathogens and antibiotic resistance [3,38]. Langford et al. [39] reported that calves fed with different penicillin residues in waste milk developed antibiotic resistance of gut bacteria. In the study conducted by Zou et al. [40], waste milk resulted in different degrees of enteritis and diarrhea in calves. In addition, the efficacy of pasteurization and acidification has been described to limit bacterial load and prevent diarrhea [40,41]. In our study, feeding with milk replacer also showed greater odds of diarrhea. Godden et al. [42] noted lower morbidity and mortality rates among calves fed pasteurized waste milk compared to milk replacer. The authors explained their results considering the impact of nutrition on immune function and that milk possibly contains other beneficial components that milk replacer does not, such as immunoglobulins and other immune factors. 

Despite a large amount of existing evidence about the disadvantages of conventional feeding programs (restrictive feeding), it is still widely used on dairy farms in Chile. Restrictive feeding consists of providing 10% of body weight in terms of milk (4 to 5 L/day); the intensified feeding programs include feeding calves with more milk (6 L/day up to ad libitum) [43]. Jasper and Weary [43] mentioned that conventional feeding is used due to the perception that increasing milk consumption increases diarrhea problems; also, it is associated with less solid feed consumption [44,45,46]. Although smaller quantities of milk encourage rumen development due to more solid feed consumption, this becomes relevant from the third week of age, so the limited amount of milk before that period is not justified [46]. Even so, in many studies, the post-weaning solid feed consumption was similar between high and low milk regimes, always showing greater weight gains and productive indices in those calves fed with more milk [9,45,47]. Besides, calves fed with less milk (4 L/day) show signs of chronic hunger, such as visiting the feeder more times, being more competitive, and resting less [9,45,47].

In our study, most farms did not offer water to calves in the first week. Vasseur et al. [3] described that 10% of the surveyed farms provided water for the first time after weaning, and the average age reported was 2.5 days. Water is an essential requirement that must be granted from the early days of life [10]. Farmers must consider giving water from birth, as restrictive milk feeding regimes result in increased water ingesting [48]; also, water stimulates the consumption of solid feed and therefore, has an impact on weight gain [41]. 

We found several farms that gave starter and hay for the first time 30 days after birth. Vasseur et al. [3] reported an average of seven days for offering concentrate and three days for hay. Peli et al. (2016) [49] reported that a critical aspect of dairy farms in Italy is the lack of ration of fibrous food in calves over two weeks. The consumption of solid feed is necessary for proper rumen development [10,46]. According to Drackley [50], consumption of starter stimulates the differentiation of the ruminal epithelium and changes in the microbial population. Although hay consumption may be a contentious issue [50], hay is important for ruminal development and encourages total dry matter consumption [51]. 

#### 4.1.5. Weaning

In the current study, most of the farms used age as a criterion for weaning—average age 80 days. Vasseur et al. [3] found an average of 49 days, and Santos et al. [27] reported a weaning age between 90 and 150 days. Weaning is a stressful process, especially considering that under commercial conditions, calves are weaned at younger ages compared to the natural process [3,5]. De Passillé et al. [48] stated that delaying weaning from 47 to 89 days reduced the drop in energy consumption and the behavioral signs of hunger. 

#### 4.1.6. Painful Procedures

Similar to Vasseur et al. [3], the average age for disbudding was 45 days. Regardless of age, disbudding is a highly painful procedure [52,53,54], and it is performed routinely without the benefit of pain control. Most farms in our study did not use analgesia or anesthesia. Des Roches et al. (2014) [55] described that only 24%, 2.3%, and 2.3% of surveyed farms used anesthetics, analgesics, or both during disbudding in French dairy farms. Local anesthesia (such as lidocaine) lasts 2–3 hours post disbudding; consequently, the joint use of local anesthesia and analgesics (NSAIDs) is recommended to maintain stable cortisol levels and reduce behavior associated with pain [52,56]. In the present study, only two farms considered both analgesia and anesthesia during disbudding.

Animals that suffer injury, disease, or are considered dispensable for the farm should be humanely killed; hence, a euthanasia program is essential [10]. In the current study, most farms did not perform euthanasia, which means that calves were bled without previous stunning or died in the pens by natural death. Other studies reported high rates of unassisted deaths; Roche et al. [57] informed that 89% of mortality in preweaned heifers among Canadian farms corresponded to unassisted deaths. Although Chilean law promotes the use of humane slaughter of farm animals [58], calves continue to be killed unacceptably on many farms. The reasons for this were not deepened in this study. However, in informal conversations with calf managers, they preferred not to kill the animals because they wanted to give them the chance to recover from illness or the situation was conflictive because they had become very close to their animals. According to the literature review by Shearer [59], euthanasia of animals entails moral stress and emotional conflicts for the people involved. What is highlighted in this study is the lack of knowledge about euthanasia, which can lead to poor decision-making, undesirable practices [60], and breakage of the law. It is significant to mention that we did not investigate the pharmacological methods used on the farms that reported using it; therefore, we cannot conclude that these farms are performing acceptable practices of euthanasia.

### 4.2. Facility-Based Measurements

Raising dairy calves in group pens is based on the principle that they are herd animals. In the current study, we found that most farms housed their calves in group pens from birth. On the contrary, Vasseur et al. [3] found that 87% of the farms kept calves individually. One of the reasons many farmers adopt individual hutches is because social group housing can have negative effects on health and productive performance [61]. However, isolation during early age is associated with adverse effects. Costa et al. [62] suggested that calves housed individually have poor social skills, are less apt to face new events, and have lower learning abilities. On the other hand, calves raised with partners improve their solid feed consumption, increase weight gain, and begin to ruminate earlier [11,62,63]. 

As in previous studies [3], most farms used straw as bedding material. Tuyttens [64] concluded that straw brings benefits such as reducing stereotypes and increasing rumination and feed consumption. We found several pens with an insufficient depth of straw. Peli et al. (2016) reported that 14% of surveyed farms in Italy did not provide appropriate bedding for calves less than two weeks old. The amount of bed material is a reflection of nesting ability, which has implications for calf health. Lago et al. [65] reported that the prevalence of calf respiratory disease decreased by increasing nesting score (legs not visible when lying). 

Although only two farms used elevated pens with slotted wood flooring for waste drainage, this system with no bed material can bring negative consequences such as lameness [10]. In the present study, 40% of the lame calves belonged to one of the farms with this type of housing. 

In this study, less space allowance in the pen (≤1.8 m^2^/calf) increased the odds of respiratory disease [OR = 2.13; *p*_wald_ < 0.05; CI_95_ = 1.04, 4.36]. Our result is in disagreement with the study by Brscic et al. [66], who found that greater space allowance (>1.8 m^2^/calf) increased the risk of occurrence of nasal discharge. The authors explained that by having less space, calves reduce their activity; therefore, there is less dust in the environment, which can be detrimental to the respiratory tract. However, other studies describe that less space is associated with poor health and performance indicators [67,68]. According to Calvo-Lorenzo et al. [67], calves housed in individual hutches with greater space (3.71 m^2^/calf) exhibited better pulmonary immunity, showing less eosinophil infiltration. Besides, an increase in the available space is associated with more active animals; it gives the possibility to display play behavior, and decreases non-nutritive oral behaviors [69,70,71]. Additionally, in pens with more space, the transmission of pathogens could be more limited. All these factors may have contributed to improving the calves’ health performance.

In the present study, the frequency of bed cleaning was related to diarrhea. Cleaning the calf’s bed 2–3 times/week [OR = 4.94; *p*_wald_ < 0.01; CI_95_ = 1.85, 13.14] or once/week [OR = 11.89; *p*_wald_ < 0.01; CI_95_ = 3.54, 39.97] were associated with increased odds of diarrhea (compared with every day). This result may represent the hygienic conditions in which calves are kept. Previous studies conclude that cleaning is an important management factor in preventing high levels of diarrhea agents in calves [72,73]. For example, Mohammed et al. [72] found that daily cleaning of the calf’s bed (either by removing only the soil, all bedding material, or partial cleaning) was associated with a lower risk of infection with *Cryptosporidium parvum* in unweaned calves.

### 4.3. Animal-Based Measurements

We found a high prevalence of sick animals (with at least one health problem) with high variability among farms. It is essential to consider that the prevalence of all health problems reported in the present study may be underestimated, due to the small number of calves evaluated on some farms and the fact that diagnosis of the diseases was performed only once by visual inspection. What is important to highlight is that a fundamental requirement to maintain a good level of animal welfare is to keep animals healthy; this includes identifying sick animals, keeping records, and preventing diseases [10]. In the current study, respiratory problems and diarrhea showed a low prevalence compared to other publications [13,14]. For example, Medrano-Galarza et al. [14] reported pen-level prevalence on farms with automated milk feeders of 23% and 17% for diarrhea and respiratory problems, respectively. 

We found that calf cleanliness score was associated with diarrhea and respiratory disease. Moderately dirty calves [OR = 5.31; *p*_wald_ < 0.001; CI_95_ = 2.50, 11.27] were associated with increased odds of diarrhea. Likewise, moderately dirty [OR= 3.25; *p*_wald_ < 0.05; CI_95_ = 1.73, 6.09] and dirty calves [OR = 4.95; *p*_wald_ < 0.05; CI_95_ = 1.68, 14.53] had higher odds of having respiratory disease. These results should be interpreted with caution because only one evaluation was performed; thus, it cannot be concluded whether the calf cleanliness score is a cause or an effect of the health condition. Even so, this may reflect the level of cleanliness and comfort of the resting areas in pens [10]. Quality of bedding has been related to health aspects. Medrano-Galarza et al. [14] found a protective effect of frequently adding fresh bed material in the prevalence of diarrhea. They also reported that in pens with more wet bedding packs, the prevalence of respiratory disease increased.

The findings of this study have to be seen in light of some limitations. First, the number of recruited farms was restricted due to time and budget constraints, particularly considering that only one farm could be sampled per day and that the farms were 50 to 100 km apart from each other; however, the sample of farms represents the local reality in Los Ríos region. Second, the target population was not randomly selected; we tried to limit the bias of our sampling strategy by choosing farms based on selection criteria that were not related to any of the variables of interest. Another limitation was that causal inferences could not be made because only one visit could be scheduled, so we were unable to follow the animals. The clinical scoring assessment was another limitation: by performing only one visual clinical assessment of diarrhea and respiratory disease, the accuracy of diagnosis was limited; thus, probably the number of sick calves per farm is also underestimated.

## 5. Conclusions

Overall, the results of the present study suggest that despite the vast amount of research available on health, productivity, and welfare of dairy calves worldwide, many Chilean farmers are still performing management practices that could be detrimental for the welfare of their animals. The recommendations that could be extrapolated from this study are that refining management practices, such as checking colostrum quality, avoiding the use of waste milk, giving more space in the pens, and maintaining a good quality of bedding, may help farmers to improve the health of their calves. Additionally, we found several potential welfare risk factors such as a high percentage of farmers that perform disbudding with no pain control, and lack of euthanasia protocols for animals that need to be killed/slaughtered. Although this study focused on describing critical points, we also found remarkable positive aspects that must be considered as a breakthrough in the matter, such as the increasing use of colostrometer or refractometer to measure colostrum quality and passive transfer evaluations. The type of approach in this study allows us to discriminate where intervention strategies should be focused in order to improve the health and welfare of dairy calves in Chile.

## Figures and Tables

**Table 1 animals-10-01115-t001:** Calf health scoring criteria to categorize calves according to its health status on 29 dairy farms using group pens in Los Ríos region, Chile.

Health Factor	Scoring System			
	0	1	2	3
Body condition ^1^	Normal: the calf is of the same weight and condition as the average of the batch	Lower condition: the calf is between 15 and 30% below the average of the batch	Severe lower condition: the calf is 30% below the average weight or condition of the batch	-
Calf cleanliness score ^1^	Clean: no manure	Moderately dirty: less than 25% of the surface is covered by manure	Dirty: More than 25% of the surface is covered by manure	-
Fecal score ^2^	Normal	Semi-formed, pasty	Loose, but stays on top of bedding	Watery, sifts through bedding
Eye score ^2^	Normal	Small amount of ocular discharge	Moderate amount of bilateral discharge	Heavy ocular discharge
Cough ^2^	None	Induce single cough	Induced repeated coughs or occasional spontaneous cough	Repeated spontaneous coughs
Skin lesion ^1^	No evidence of spots of skin lesions	Evidence of spots of skin lesions.	-	-
Lameness ^1^	No evidence of lameness	Evidence of lameness	-	-

^1^ European Welfare Quality^®^ protocol. ^2^ Protocol of the University of Wisconsin-Madison.

**Table 2 animals-10-01115-t002:** Protocol-, facility-, and animal-based measurements gathered through interviews with calf managers, visual inspection by the researcher, and farm records during visits to 29 dairy farms using group pens in Los Ríos region, Chile.

Measurement and Variable	Source of Data	Scale	Score/Average
**Protocol-based Measurements**			
N° calves in the calf barn	Interview	Continuous	165 ± 136 calves
Main breed on the farm	Interview	0: Holstein; 1: dairyXbeef; 2: HolsteinXJersey	0: 12; 1: 8; 2: 9 farms
Presence of maternity pen	Interview	0: yes; 1: no	0: 19; 1: 10 farms
Supervision during calving	Interview	0: yes; 1: no	0: 27; 1: 2 farms
Time mother-calf spent together	Interview	Continuous	23.12 ± 31.95 hours
Colostrum source	Interview	0: manual; 1: mother	0: 15; 1: 14 farms
Evaluation colostrum quality	Interview	0: yes or 1: no	0: 9; 1: 20 farms
Evaluation passive transfer	Interview	0: yes or 1: no	0: 8; 1: 21 farms
Type of milk	Interview	0: milk replacer; 1: treated waste milk; 2: untreated waste milk	0: 8; 1: 6; 2:15 farms
Liters of milk	Interview	0: ≤4 L/day; 1: >4 L/day	0: 21; 1: 8 farms
Milk feeding frequency	Interview	0: 2 times/day; 1: 4 times/day	0: 28; 1: 1 farms
Age of access to water	Interview	Continuous	10.3 ± 15.5 days
Age of access to starter	Interview	Continuous	6.3 ± 8.3 days
Age of access to hay	Interview	Continuous	14.9 ± 16.9 days
Weaning criteria	Interview	0: age; 1: weight	0: 25; 1: 4 farms
Weaning age	Interview	Continuous	79.8 ± 18.0 days
Weaning weight	Interview	Continuous	79.4 ± 14.8 kg
Disbudding: age	Interview	Continuous	44.9 ± 28.3 days
Disbudding: analgesic	Interview	0: yes; 1: no	0: 3; 1: 26 farms
Disbudding: anaesthetic	Interview	0: yes; 1: no	0: 6; 1: 23 farms
Euthanasia protocol	Interview	0: yes; 1: no	0:10; 1:16 farms
**Facility-based measurements**			
Time in individual hutches	Interview	Continuous	6.4 ± 4.7 days
Bedding material	Inspection	0: straw; 2: pasture; 3: without bedding material	0: 25; 1: 2; 2: 2 farms
Bed material depth	Inspection	Continuous	7.9 ± 4.4 cm
Space allowance	Inspection	0: ≤1.8 m^2^/calf; 1: >1.8 m^2^/calf	0: 23; 1: 56 pens
Bed cleaning frequency	Inspection	0: every day; 1: 2–3 times/week; 2: 1 time/week	0: 12; 1: 11; 2: 4 farms
**Animal-based measurements**			
Body condition score	Inspection	0: normal; 1: lower condition; 2: severe lower condition	0: 596; 1: 89; 2: 15 calves
Calf cleanliness score	Inspection	0: clean; 1: moderately dirty; 2: dirty	0: 494; 1: 176; 1: 23 calves
Season of birth	Record	1:autumn; 2: summer; 3: spring; 4: winter	1: 227; 2: 178; 3: 143; 4: 152 calves
Sex	Record	1: female; 2: male	1: 443; 2: 252 calves
Age	Record	Continuous	45 ± 20 days

**Table 3 animals-10-01115-t003:** Calf and herd-level prevalence (%) of respiratory disease, diarrhea, and other health problems on 29 dairy farms using group pens in Los Ríos region, Chile.

Health Event	Calf-Level Prevalence			Herd-Level Prevalence					
	Total Calves Assessed	Cases	Overall ^1^ (%)	Mean Herd Prevalence ^2^ (%)	Min	1st Quartile	Mean	3rd Quartile	Max
Respiratory disease	700	109	15.5	17.7	0	10.0	14.3	21.4	75.0
Diarrhea	686	94	13.7	12.7	0	3.3	6.7	21.7	41.7
Other health problems ^3^	700	104	14.9	15.3	0	2.9	12.5	21.7	75.0
Sick ^4^	700	260	37.2	38.1	3.1	26.7	35.7	47.8	83.3

^1^ Overall (%) = n calves with a health problem/n calves examined. ^2^ Mean herd prevalence (%) = n calves with a health problem/n calves examined. Divided into mean herd prevalence, minimum, 1st quartile, mean, 3rd quartile, and maximum. ^3^ Other health problems—lameness, skin lesions, blindness, umbilical hernia, lumpy jaw, and evidently ill animals. ^4^ Sick—percentage of calves with at least one health problem (respiratory disease, diarrhea, and other health problems).

**Table 4 animals-10-01115-t004:** Final multilevel regression model for diarrhea in 686 ^1^ calves raised in group pens on 29 farms in Los Ríos region, Chile.

Factor	Level	Estimate	SE	OR	95% Wald CI for OR	*p*-Value	Overall *p* Value
Intercept		−5.04	0.80				
Number of calves in the calf barn	≤71	−2.45	0.69	0.85	0.02–0.33	0.0005	0.002
	>71 to ≤129	−1.29	0.70	0.27	0.06–1.09	0.66	
	>129 to ≤215	−2.31	0.74	0.09	0.02–0.43	0.002	
	>215 (ref)						
Frequency of bed cleaning	2–3 times/week	1.59	0.49	4.94	1.85–13.14	0.001	0.0002
	Once/week	2.47	0.61	11.89	3.54–39.97	<0.0001	
	Every day (ref)						
Calf cleanliness score	Moderately dirty	1.67	0.38	5.31	2.50–11.27	<0.0001	<0.0001
	Dirty	1.17	0.70	3.22	0.80–12–91	0.09	
	Clean (ref)						
Milk type	Milk replacer	2.14	1.00	8.56	1.18–61.94	0.03	<0.0001
	Waste milk	3.43	0.86	31.02	5.65–170.21	<0.0001	
	Treated milk (ref)						

^1^ Fourteen calves could not be assessed for diarrhea.

**Table 5 animals-10-01115-t005:** Final multilevel regression model for respiratory disease in 700 calves raised in group pens on 29 farms in Los Ríos region, Chile.

Factor	Level	Estimate	SE	OR	95% Wald CI for OR	*p*-Value	Overall *p* Value
Intercept		−2.59	0.52				
Colostrum quality evaluation	No	0.84	0.42	2.28	1.01–5.13	0.04	
	Yes (ref)						
Calf cleanliness score	Moderately dirty	1.18	0.31	3.25	1.73–6.09	0.0002	0.0003
	Dirty	1.59	0.54	4.95	1.68–14.53	0.003	
	Clean (ref)						
Space allowance	≤1.8 m^2^/calf	0.75	0.36	2.13	1.04–4.36	0.03	
	>1.8 m^2^/calf (ref)						
Frequency of bed cleaning	2–3 times/week	−0.007	0.43	0.99	0.43–2.32	0.98	0.64
	Once/week	0.46	0.55	1.58	0.53–4.76	0.40	
	Every day (ref)

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
