# Peer review of "Dairy Calf Welfare and Factors Associated with Diarrhea and Respiratory Disease among Chilean Dairy Farms"

_animals, 2020, doi:10.3390/ani10071115_

Round 1

Reviewer 1 Report

This study used both indirect and direct measures to evaluate dairy calf welfare in Chile. This manuscript gives insight into dairy calf management, health, and welfare in Chile. This is an important area of study in order to progress global animal welfare. The statistical analysis of diarrhea and respiratory disease predictors was a beneficial addition to the paper. I do not have any additional suggestions for this manuscript. 

Reviewer 2 Report

Review of Animals 813634

General Comments

The manuscript “Dairy calf welfare and factors associated with diarrhea and respiratory disease among Chilean dairy farms” describes the results of a survey and animal and facility observations on 29 farms. The reviewed version has highlighted text that appears to have been added based on a previous review. Calf welfare is an important topic and this paper does have good information relative to the Chilean dairy industry. Unfortunately, the incorporation of associations with diarrhea and respiratory disease, although interesting and important, aren’t well weaved into the paper. There are also many grammatical errors that make some sections difficult to follow the line of thought. I believe the paper has merit but needs some additional work.

Specific Comments

Line 29 – I could not find the definition of treated milk. What were the milk replacer formulations?

Line 30 – was this a standardized cleanliness score? If so please reference. Maybe refer to as calf cleanliness to differentiate from housing cleanliness? When bed cleaning and cleanliness score are together in the table, it could be misinterpreted. Same in table 5.

Line 63 – these are important calf diseases, some additional information would be to report incidence estimates and associations found from other studies.

Line 138 – was BCS scoring standardized? Reference?

Line 155 – was the higher p value variable retained or removed?

Line 169 – two decimal places for percentages on 29 farms seems excessive. Does Animals require that precision?

Table 2 – Season instead of date of birth

               Why do some responses not add the 29? Colostrum grant (source) only adds to 19 for example

Euthanasia protocol is either yes or no. Bleeding without stunning is killing, not euthanasia and unassisted death is also not euthanasia. It appears that two points are captured here and the title doesn’t work.

               Does bed cleaning frequency relate to frequency of adding bedding as opposed to cleaning?

Table 3 – positive cases is redundant, suggest removing positive

               Some values have one decimal point and others have two.

Line 213 – interesting that evaluation of colostrum quality was associated with resp disease but not passive measures. Were measurement of colostrum quality and passive immunity correlated?

Line 337 – should breeding be housing? Same on line 339

Line 344 – when referring to ‘higher’ was any comparison of confidence interval included? 51 might not be higher than 48 statistically.

Reviewer 3 Report

The topic of the paper is of huge interest and the Authors addressed it presenting the results of a survey performed on 29 dairy farms.

This number is the main limit of this survey. In fact, the sample dimension is not statistically supported and the selection criteria adopted might have biased the survey (“All participant farms were convenience sampled by means of personal contacts with the industry”). Besides, the size range of the farms appears to be too wide to obtain unbiased data, unless it might be possible to analyze the results on the basis of this factor analizing how the farm size might influence the observed phenomena.

The authors are invited to discuss and comment on these critical aspects in order to support the scientific value of their results.

A second point that is worth to be remarked and deeper analyzed is the correlation between the  ABMs and the age of calves. In fact, it is well known that the gastrointestinal disorder as well as the respiratory diseases have  different prevalence depending on the age of the calves and, in particular, the younger calves (<30 days of age) are at highest risk. Sex has also to be considered as an important factor that may affect the prevalence of these (and others) calves’ diseases because of different commercial destination of the animals (i.e. fattening farm or remount to replace dairy cows) which, in turn, may affect the level of care given by the farmers during the first neonatal period.

The authors are invited to analyze and discuss their results on diarrhea and respiratory diseases in the light of these factors.

The Authors are invited to analyze their results considering the effect of the season in particular on the incidence of diarrhea and respiratory diseases.

Furthermore, it would be interesting to know the mortality rate of calves recorded in each farm (overall; < 30 days; > 30 days of age). This is known to be a very sensitive parameter of welfare status, although provided with low specificity.

Did the Authors record data about colostrum bank besides colostrum quality measurement? If not, why?

The use of oesophageal tube and the use of the bottle cannot be considered as equivalent methods in terms of animal welfare and efficacy to obtain a satisfactory level of passive immunity. Please comment on this aspect with regard to the results obtained.

Regarding the disbudding, please note that, in accordance with EU legislation and EFSA recommendations, it should be done within 21 days of age and, consequently, it appears that all the farms interviewed are nonconforming.

It would be interesting to know the percentage of farms practicing the disbudding.

The number of staff and its ability, knowledge and professional competence are factors that can strongly impact on animal welfare and it is important to include these data on the analysis.

It would be also recommended to compare the results presented in this paper to the ones obtained from extensive, recent surveys on bovine welfare carried out in other countries.

Row 47 I suggest changing the reference n. 1 because the definition was originally done by

Broom D.M., 1986. Indicators of poor welfare. British Veterinary Journal, 142, 524-526.

The reference n 18 is not cited along with the text but only in the references section; furthermore, itis not correct since the cited directive had been repealed by Dir 2008/119/EC.

Round 2

Reviewer 2 Report

The authors have addressed all of my concerns.

Author Response

Your comments and suggestions greatly improved our manuscript. Thank you.

Reviewer 3 Report

Point 1 - The Authors are kindly invited to demonstrate how they can define “adequate” and representative the sample size they adopted. This assumption should be supported by a power analysis or, at least, by a comparison to the number of farms as well as of mean calves’ population of the region were the survey was carried out.

Point 9 -The following references may be suggested in order to help comparing and discussing the results obtained as well as supporting the justification of the sample dimension of the survey.

  1. Jožica Ježek et al. (2020): “Management practices affecting calves welfare on farms in Slovenia”. Acta Univ. Agric. Silvic. Mendelianae Brun. 2019, 67, 1147-1152
  2. Reimus, K., et al. “Herd-level risk factors for cow and calf on-farm mortality in Estonian dairy herds”. Acta Vet Scand 62, 15 (2020).
  3. Peli A., et al. (2016): “Survey on animal welfare in 943 Italian dairy farms”. Italian Journal of Food Safety, 5, 50-56. doi: http://dx.doi.org/10.4081/ijfs.2016.5832
  4. des Roches, A de Boyer et al. (2014): “The major welfare problems of dairy cows in French commercial farms: an epidemiological approach”. Animal Welfare, 23, 4, 467-478

Author Response

This manuscript is a resubmission of an earlier submission. The following is a list of the peer review reports and author responses from that submission.

Round 1

Reviewer 1 Report

This study used both indirect and direct measures to evaluate dairy calf welfare in Chile. The manuscript is well written and gives insight into dairy calf management, health, and welfare in Chile. This is an important area of study in order to progress global animal welfare. The biggest drawback I saw was the lack of statistical analysis of any of the measures. While descriptive statistics seem adequate for reporting data on industry norms, this manuscript would be greatly improved by running statistical analyses of group differences (for example differences between sex, breeds, and ages). Below are specific recommendations for the manuscript:

L114-117 I recommend changing the wording of the calf score (fearful, cautious, friendly) to be less subjective (e.g. not approachable, moderately approachable, approachable) here and throughout the manuscript. As the authors explained in the discussion, a calf that is approachable may not always be friendly (i.e. could also be curious or sick). 

L169 (Table 2) A statistical analysis should be run on this data (at least a t-test comparing heifer and bull differences). This would help support claims made in the conclusion. 

L185 A written explanation should still be given of these results

L198 (Table 4) A statistical analysis should be run on this data

L219-223 It would be nice to flesh this out a bit more. Since the average time in this study was ~25 hours, would these producers still see the benefits noted in the cited studies? Was there a commonly noted reason for leaving calf and cow together for this length of time? 

L280-283 I would like to see more discussion on this as it's a major finding of the study. For example, are there any animal euthanasia laws that could impact these results? Were there any commonly noted reasons for lack of euthanasia standards such as lack of training, lack of veterinary availability/oversight, lack of staffing, commonly held beliefs about animal suffering, etc.? 

Reviewer 2 Report

This is study will benefit the Chilean dairy industry as it points out potential welfare issues that calves are be facing. However, the manuscript in its current format has some problems:

- Major: descriptive statistics considered the calf as the experimental unit instead of the farm. Calves are not independent in each farm, so they should be considered as so in the data analysis. This section should be re-done so we can truly evaluate the results, discussion, and conclusions.

- Minor: the flow of the manuscript could be improved as well as more background information should be provided so the reader can better understand what has been done and why. Also, the terminology could be improved using terms that are more frequently used in the welfare evaluation/assessment field to facilitate comparison. In addition, some information presented as a welfare "problem" is outdated and should follow the current literature (e.g., disbudding).

Please see the specific comments for more details.

Specific comments

L2– I would rather use “a description” because it reflects what it was done. “View” reflects thoughts about something.

L11 to 22– simple summary should inform non-academic readers. As of now, I do not think this is being accomplished. For example, what does it mean to measure 736 calves? Also, what does it mean to measure behavior?

L23 – the introductory sentence is vague. I think L11 described better your problem than this one.

L24, 25 – considering that there are many different rearing systems for calves, I would like to have some idea here about the type that you evaluated (i.e., when they were grouped, age 46 d).  Also, isn’t clear the number of calves per farm and your sample size (per farm). My understanding is that you did not sample all 31 farms for all measures – so you need to provide the n  (number of farms) for each measure.

L25,26 – to be more consistent with AW evaluations I would use the terms: facility-based, protocol-based and animal-based measures and give a brief description.

L27 – I would use the word “likely” or “potentially” identified for poor welfare if you aim to keep this sentence. You do not provide data to show these practices cause poor welfare in the animals evaluated. I rather just present the results rather than conclude anything here – as you are only describing practices.

L31 – A study published last year (Adcock & Tucker 2018, Journal of Dairy Science) showed no effect of age on healing/pain after disbudding. There is no other scientific evidence that shows otherwise.

L34, 36 – your conclusions sound just like your introduction. Please summarize your results: what are the most important aspects for calf welfare that need to be addressed in Chile?

L47, 48 – should clarify that the Welfare Quality refers to Europe. Also, I would add a statement saying that this is how they evaluate welfare in practice to clarify your measurements.

L49,51 – Instead of stating what is “wrong” I would define what is the best practice. For example, when you talk about good nutrition, the best practice is to give milk ad libidum (which is about 20% of their BW). So the reader can better understand why feeding 10% of their BW is not appropriate. The same applied to all the examples given in this paragraph.

L65, 68 – I think the main objective should describe the calf rearing systems in Chile because you currently, in Chile, do not have any systematic information?

L71 – what does it mean to measure dairies in Los Rios in Chile? Is this the largest dairy region? Are they pasture-based, free-stalls? More detail is needed in this paragraph to understand the context.

L78 – to complete data collection.

L79,80 – consider my comments for the abstract about how to name the measurements taken. Also, I would re-do table 1 adding all measurements taken splitting them by category (protocol-, facility-, and animal-based). You could only star the ones you collected by interview.

L82 – please attach the whole questionnaire as an appendix

L89 – 1.58 m tall. What about intra-observer reliability?

L89-91 – The differences in number were caused by non-completion of calves per farm or lack of collection in some farms? My understanding is that you did not sample all 31 farms for all measures – so you need to provide the n  (number of farms) for each measure. As now it seems that your sample size is that big but actually your sample size is the number of farms sampled. Please make that clear throughout the manuscript.

L102 – from outside the pen? Please provide more details. About the observations.

L104 – lumpy jaw

L105-106 – Please refer to UW-Madison protocol on your references to match.

L107 – I am confused about the “unfamiliar person”. My understanding is that you measured health first, then their reaction to your presence? If so, that does not make much sense as the calves could have habituated. Also, can you please provide more information about the validity of this test? As now I am not convinced you can use it to determine fear.

L114-115 – I need to see more data here to have a better understanding that lack of eye contact means fear. To me it can mean so many things: they can be distracted with something else; they can be shy, and so on.

L117-118 – I am concern about the validity of the 2 farms that were scored before feeding. I assume calves are more likely to approach before feeding, especially in this situation where they seem to be so hungry. You have a fair amount of data, so I would drop it to keep it consistent.

L119-125 – What was the average age of the second weight for the 10 farms that measured calves at birth? Also, I think you should treat differently the 2 methods of data collection; unless you have literature supporting that they are the same. For me, it does not make sense that ADG will be the same during day 25-58, than from birth to any age because their management is very different (e.g., water provision). Thus, they should be considered different.

L127-133 – Even the results are descriptive, all measures should be calculated by FARM, which is the experimental unit. As now the results are pseudo-replicated, a calf within a farm cannot be considered independent.

Round 2

Reviewer 2 Report

Thank you for sending an updated version of this MS. The content is better organized, thus the flow is improved but the MS should be carefully examined for grammar. Still, as said on my review #1, I disagree on how you approached data analyses for the animal-based measures. Keeping the calf as experimental unit (instead of farm) results in pseudo-replication. Animal-based outcomes can be a direct product of management, thus it is applied to the farm level. When you consider calf as the experimental unit, you can be over or underestimating the problem, so your conclusions would be compromised.

A brief example: you visit 10 farms and score sickness in 10 calves/farm. At total you found 10 sick calves. However, if you do not know the distribution of sick calves/farm you cannot draw any conclusions because having 10 calves sick at the same farm is different than having 1 calf sick in each farm or a couple sick in every other farm. If all the sick calves are in the same farm, this unlikely will be a welfare concern for Chilean calves. This likely will be a welfare concern for the calves housed in that specific farm.
